# Precise regulation of the guidance receptor DMA-1 by KPC-1/Furin instructs dendritic branching decisions

**Xintong Dong[1,2], Hui Chiu[3†], Yeonhee Jenny Park[4], Wei Zou[1,2], Yan Zou[3‡], Engin Özkan[4], Chieh Chang[3*], Kang Shen[1,2]***

[1]Department of Biology, Stanford University, Stanford, United States; [2]Howard Hughes Medical Institute, Stanford University, Stanford, United States; [3]Department of Biological Sciences, University of Illinois at Chicago, Chicago, United States; [4]Department of Biochemistry and Molecular Biology, University of Chicago, Chicago, United States

**Abstract** Extracellular adhesion molecules and their neuronal receptors guide the growth and branching of axons and dendrites. Growth cones are attracted to intermediate targets, but they must switch their response upon arrival so that they can move away and complete the next stage of growth. Here, we show that KPC-1, a *C. elegans* Furin homolog, regulates the level of the branching receptor DMA-1 on dendrites by targeting it to late endosomes. In *kpc-1* mutants, the level of DMA-1 is abnormally high on dendrites, resulting in trapping of dendrites at locations where a high level of the cognate ligand, the adhesion molecule SAX-7/L1, is present. The misregulation of DMA-1 also causes dendritic self-avoidance defects. Thus, precise regulation of guidance receptors creates flexibility of responses to guidance signals and is critical for neuronal morphogenesis.

***For correspondence:** chiehc@uic.edu (CC); kangshen@stanford.edu (KS)

**Present address:** [†]Division of Biology and Biological Engineering, California Institute of Technology, Pasadena, United States; [‡]School of Life Science, Shanghai Tech University, Shanghai, China

## Introduction

Developing neuronal dendrites and axons navigate through complex environments before establishing their final morphologies (*Tessier-Lavigne and Goodman, 1996*; *Dickson, 2002*; *Jan and Jan, 2010*). Extracellular cues provide neurites with instructive spatial signals to guide their growth, turning and branching decisions. Different neurons can interpret the same cue differently based on the receptors they express. Many guidance cues support both attractive and repulsive responses, which are mediated by different classes of receptors (*Tessier-Lavigne and Goodman, 1996*). Axon guidance is achieved not only through cell-specific expression of receptors but also by dynamic regulation of those receptors during neural development.

A classical example of spatial and temporal regulation of guidance receptors is midline crossing in the *Drosophila melanogaster* central nervous system (CNS). In the fly nerve cord, many neurons extend their axons across the midline to the contralateral side while others remain on the ipsilateral side (*Harris et al., 1996*; *Kolodziej et al., 1996*; *Mitchell et al., 1996*). This process critically depends on the midline repellent Slit, its receptor Roundabout (Robo) and another protein Commissureless (Comm) (*Kidd et al., 1998a*; *1998b*; *1999*; *Brose et al., 1999*; *Wang et al., 1999*). Slit is secreted by midline cells and repels growth cones expressing the Robo receptor. Longitudinal axons that do not cross contain high levels of Robo and are kept away from the midline (*Kidd et al., 1999*). Comm, on the other hand, functions in the contralaterally projecting commissural neurons to keep the level of Robo low before commissural axons cross the midline, during which the growth cones need to ignore the repellent Slit. The expression of Comm diminishes after growth cones

**eLife digest** Neurons are the principal cells in the nervous system that send and receive information. A vast network of neurons helps transmit information throughout the brain and body. The end of the neuron that receives messages forms branched structures called dendrites, the shapes of which determine the signals the neuron receives. Therefore, establishing the correct shape of the dendrites is critical for the neurons to work correctly.

As dendrites grow during development, signals from the environment tell them where to branch and where to stop. For example, the neurons that transmit information about touch respond to signals from skin cells to guide the growth of their dendrites. These signals bind to receptor proteins on the surface of the neuron. However, the environment around the neurons also contains many guidance signals that the neurons must ignore.

Dong et al. now show that touch neurons control how they respond to signals by adjusting the abundance of the receptors on their surface. First, genetic mutations were identified that distort the shape of the dendrites of touch-sensing neurons in a simple worm called *Caenorhabditis elegans*. These neurons lacked the equivalent of an enzyme called Furin and had abnormally high amounts of a receptor protein called DMA-1 on their surfaces. This suggests that controlling the receptor level on dendrites creates flexibility in the guidance choices of dendrites.

Furin usually cuts up proteins. However, Dong et al. found that Furin prevents DMA-1 from inserting into the membrane of neurons by binding to the receptors and sending them to the lysosomes, cellular compartments where proteins are destroyed. Reducing the number of receptors at the surface of the cell in this way prevents the neuron from responding to the guidance signals at wrong locations. In the future, more studies are needed to understand how the neuron checks and balances this process and how it eventually is turned off.

cross the midline, allowing the Robo receptors to localize onto the plasma membrane of postcrossing growth cones so that the axons can be repelled away from the midline and prevented from recrossing (*Keleman et al., 2002*). It has been shown that Comm sorts the Robo receptors to endosomal pathways before they reach the plasma membrane (*Keleman et al., 2002*; *2005*). Similarly, in the vertebrate spinal cord, where no Comm homologue is found, differential responses to the midline repellant Slit is achieved by regulated expression and alternative splicing of the Robo3 receptor isoforms (*Sabatier et al., 2004*; *Chen et al., 2008*). Pre-crossing axons contain high levels of Robo3.1, which prevents the activation of the Robo1 and Robo2 receptors, thereby allowing axons to grow toward the highly repulsive midline. Upon crossing, Robo3.2 expression is switched on, and it acts in concert with Robo1 and Robo2 to keep axons from recrossing the midline. It remains unknown, however, how neurons achieve such spatiotemporal specificity, and the mechanisms by which Comm and Robo3 are regulated remain unclear (*Dickson and Gilestro, 2006*; *Ypsilanti et al., 2010*).

We have previously reported that dendrites of the PVD neurons in *Caenorhabditis elegans (C. elegans)* follow precisely localized guidance signals SAX-7 and MNR-1 to form highly organized dendritic structures (*Dong et al., 2013*; *Salzberg et al., 2013*). SAX-7 is the ortholog of the vertebrate neuronal adhesion molecule L1CAM (*Zallen et al., 1999*; *Chen et al., 2001*). PVD neurons first grow two longitudinally extending 1° dendrites, from which numerous 2° dendrites emerge roughly perpendicular to the 1° dendrites (*Smith et al., 2010*; *Albeg et al., 2011*). Upon reaching two sublateral lines where the guidance molecule SAX-7 is highly enriched (arrows in *Figure 1A*), the PVD dendrites form stereotyped 'T'-shaped 3° branches. SAX-7 and MNR-1 are both necessary and sufficient to instruct dendrite morphogenesis through direct interactions with the PVD receptor DMA-1 (*Liu and Shen, 2012*; *Dong et al., 2013*; *Salzberg et al., 2013*). SAX-7, MNR-1 and DMA-1 form a tripartite ligand-receptor signaling complex that enables stereotyped dendritic branch formation and stabilization. A close examination of SAX-7 localization revealed that, in addition to the sublateral stripes where 3° branches form, SAX-7 was also enriched in a zone that roughly lined up with the 1° branches (*Figure 1A*, arrowhead) (*Dong et al., 2013*). Since SAX-7 is expressed and localized well before PVD dendrite morphogenesis begins (*Dong et al., 2013*), the 2° dendrites encounter the

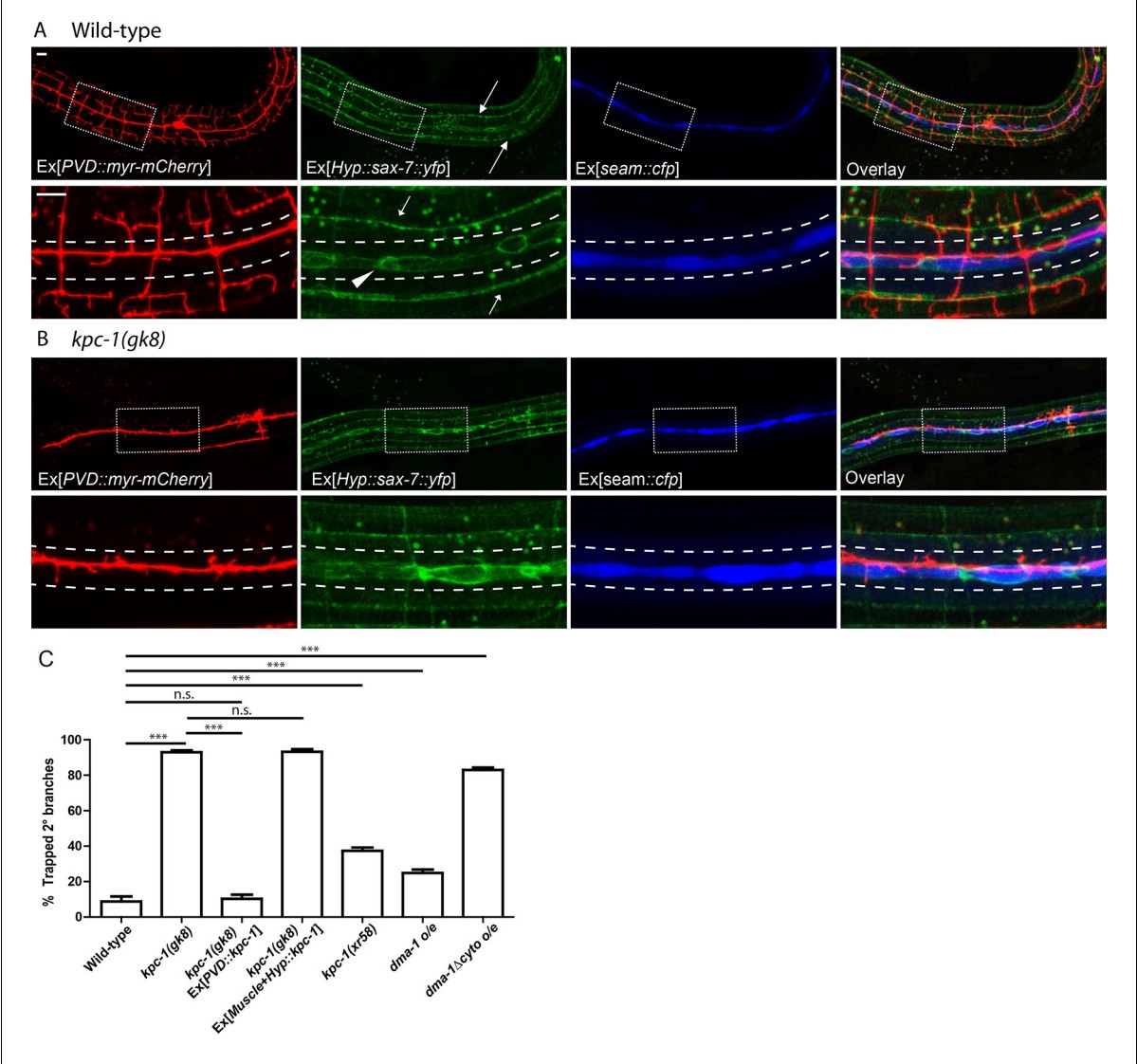

**Figure 1.** The *kpc-1* mutants showed severe trapping of PVD dendrites. (**A**) Fluorescent images showing (red) morphology of the PVD neuron, (green) localization of SAX-7 in the hypodermal cell, (blue) seam cells and overlay between the three in wild-type worms. SAX-7 was enriched in two sublateral longitudinal lines and at the lateral midline around the seam cell-hypodermal junctions. Arrows: Sublateral stripes of enriched SAX-7 that co-localize with PVD 3° dendrites. Arrowhead: SAX-7 enriched near the 1°dendrites, where it was encountered by the 2° branches as they emerge. The images in the lower panels are zoomed-in views of the regions indicated by the boxes. Dotted lines indicate the 'trap zone' marked by enriched SAX-7 around seam cells. (**B**) In *kpc-1(gk8)* mutants, almost all branches failed to grow out of the trap zone between the dotted lines indicated by enriched SAX-7. Scale bar: 10 μm. (**C**) Quantification of the percentage of 2° branches trapped around the 1°dendrite. *** is p<0.001, n.s. is p>0.05 by Student's T-test. N=50 for each genotype.

The following figure supplement is available for figure 1:

**Figure supplement 1.** Heat shock expression of KPC-1 at distinct time points during development produced different phenotypes.

SAX-7 domain as they emerge from the 1° dendrite. In wild-type animals, the vast majority of 2° dendrites grow out of the SAX-7 domain to reach the sublateral line and form 3° branches there. How emerging 2° dendrites 'escape' one attractive SAX-7 domain but grow along another SAX-7 domain is conceptually similar to how the commissural neurons dynamically adjust their axon guidance response to Slit in *Drosophila* and mouse during midline crossing.

Two recent studies have identified the worm Furin homologue KPC-1 as another key regulator of PVD dendritic development (*Schroeder et al., 2013*; *Salzberg et al., 2014*). KPC-1 is a member of the paired basic amino acid cleaving enzyme (PACE) family of proprotein convertases and is shown to be required autonomously in the PVD neurons for menorah formation. Genetic analyses showed that mutations in the SAX-7/MNR-1/DMA-1 complex were epistatic to *kpc-1* mutations (*Schroeder et al., 2013*; *Salzberg et al., 2014*). However, the functional mechanism of KPC-1 and its relationship with the receptor-ligand complex remain elusive. It is also unclear why weak hypomorphic alleles of *kpc-1* exhibit defects in self-avoidance of the 3° branches (*Schroeder et al., 2013*; *Salzberg et al., 2014*). Here, we show that KPC-1, instead of promoting branch outgrowth as proposed in the previous papers, enables the dendrites to move away from the high level of SAX-7 at intermediate targets. Our genetic, biochemical, and cell biological analyses indicate that KPC-1 supports dendrite pathfinding by maintaining a proper level of the DMA-1 receptor on the plasma membrane of dendrites. Loss of KPC-1 causes unregulated and excessive DMA-1, leading to erroneous dendrite guidance choices. These findings present a new mechanism that modulates the neuronal dendrites' response to extracellular adhesion molecules through controlling the trafficking of dendrite guidance receptors.

## Results

### Loss of KPC-1 caused PVD dendrites to be trapped around the 1° dendrite

We reported previously that one component of the PVD dendrite guidance complex, SAX-7, was enriched at specific sub-cellular locations in hypodermal cells (*Dong et al., 2013*). As shown in *Figure 1A*, YFP tagged SAX-7 expressed in the hypodermal cell was highly enriched in two longitudinal sublateral stripes (arrows) which precisely co-localized with the PVD 3° branches labeled by mCherry. In addition to these lines, SAX-7 also showed strong enrichment at ring-shaped structures that likely represented the junctions between seam cells (labeled by CFP using a seam cell-specific promoter P*nhr-81*) and the major hypodermal syncytium (*Figure 1A*, arrowhead). The seam cells are two rows of specialized hypodermal cells that line up along the lateral midlines of the worms and cover a region in which longitudinal PVD 1° dendrites extend (*Sulston and Horvitz, 1977*). The location of this SAX-7 zone dictated that PVD 2° branches would encounter an area with a high level of the adhesion molecule SAX-7 as they grow away from the 1° dendrite.

Since PVD 2° dendrites do not elaborate extensive branches in this SAX-7-rich 'trap zone', developing PVD dendrites appear to ignore SAX-7 here. We reasoned that disabling the normal escaping mechanism would lead to trapping of PVD dendrites within the zone. Indeed, in our forward genetic screen for mutants with defects in PVD morphology, we isolated three mutants, *wy916*, *wy920* and *wy936*, which showed a dramatic phenotype in which PVD dendrites were completely trapped near the 1° dendrites. This phenotype resembled that of a previously reported mutant allele *kpc-1(gk8)* (*Schroeder et al., 2013*; *Salzberg et al., 2014*). We confirmed that *wy916*, *wy920* and *wy936* were alleles of *kpc-1* by complementation test.

In contrast to the wild-type 2° dendrites, which extended away from the 1° dendrite, 2° branches in *kpc-1* mutants remained restricted to a narrow region close to the 1° dendrite (*Figure 1B*). Co-labeling of the PVD neuron, hypodermal SAX-7-YFP and the seam cells revealed that the 2° dendrites in *kpc-1* mutants were trapped inside the area with enriched SAX-7 (*Figure 1B*). We quantified the percentage of 2° dendrites that failed to extend beyond the seam cell zone near the 1° dendrite and found that the *kpc-1* mutants had a significantly larger portion of dendrites that were trapped in this zone than wild-type (*Figure 1C*). Expressing KPC-1 in PVD using the PVD-specific *ser2prom3* promoter fully rescued the dendritic defects, while expressing KPC-1 in hypodermal and muscle cells failed to rescue the phenotypes, consistent with previous reports that KPC-1 was expressed and functioned autonomously in the PVD neuron (*Figure 1C*) (*Schroeder et al., 2013*; *Salzberg et al., 2014*).

We next examined the temporal requirement for KPC-1 during dendritic development using the *hsp16.48* heat shock promoter (*Stringham et al., 1992*). Heat shock expression of KPC-1 at different developmental time points caused drastically different effects (*Figure 1—figure supplement 1D*). Expressing KPC-1 during L1 or L2 larval stages before the outgrowth PVD 2° branches did not

modify the *kpc-1(gk8)* mutant phenotype (*Figure 1—figure supplement 1A,D*). Expression of KPC-1 during the early L3 stage, in contrast, during 2° branch outgrowth, produced robust rescue of all dendritic branches: 2° branches were able to grow out of the high SAX-7 region around the seam cells and formed full menorahs (*Figure 1—figure supplement 1B,D*). Heat shock expression during later stages produced an altered phenotype in which the branches that were trapped and stabilized before KPC-1 expression remained trapped in the region while the newly developed branches could grow out and form 3° and 4° branches (*Figure 1—figure supplement 1C,D*). These results agreed with a previous report using temporal RNAi (*Salzberg et al., 2014*) and demonstrated that KPC-1 was required during a stringent time window when the 2° branches synchronously bypassed the high level SAX-7 region to allow proper outgrowth.

## The SAX-7/MNR-1/DMA-1 signaling complex mediated dendritic trapping in the *kpc-1* mutants

If the 2° dendrites in the *kpc-1* mutants were indeed 'trapped' in the region by the locally enriched SAX-7, we expected that removing the SAX-7 ligand, its cofactor MNR-1, or their cognate receptor DMA-1, would release the dendrites from the trap zone. Alternatively, if KPC-1 was required for dendritic outgrowth per se, double mutants between *kpc-1* and any member of the ligand-receptor complex would show a PVD dendritic phenotype that was similar to that of the *kpc-1* mutants. Consistent with previous reports (*Schroeder et al., 2013*; *Salzberg et al., 2014*), we observed that the *kpc-1; sax-7, kpc-1; mnr-1* and *kpc-1; dma-1* double mutants showed PVD morphologies indistinguishable from those of the *sax-7, mnr-1* or *dma-1* single mutants (*Figure 2*, *Figure 2—figure supplement 2*). Instead of remaining within a tight zone close to the 1° dendrites like in the *kpc-1* single mutants (*Figure 2A*), 2° dendrites of the double mutants extended further out in both dorsal and ventral directions (*Figure 2B*, *Figure 2—figure supplement 1*, *2*). Since the dendritic morphologies of the *sax-7, mnr-1* and *dma-1* mutants were highly disorganized with many short ectopic 2° branches, there were more 2° branches in the 'trap zone' compared with the wild-type animals (*Figure 2—figure supplement 2H*). Nevertheless, the percentages of trapped dendrites in the double mutants were comparable to those of single mutants of the tripartite complex but were significantly lower than that in the *kpc-1* single mutants (*Figure 2—figure supplement 2H*). This result suggested that defects in the *kpc-1* mutants arose as a result of ectopic activation of ligand-receptor signaling near the seam cells.

Our 'trapping' model inferred that the SAX-7 molecule was required locally near the primary dendrite to restrict 2° branches around the 1° dendrite. We next directly tested this hypothesis by ectopically expressing SAX-7 in seam cells in *kpc-1; sax-7* double mutants. In striking contrast to the double mutants themselves (*Figure 2B*), the PVD dendrites in the *kpc-1; sax-7* double mutants expressing SAX-7 in seam cells were trapped in the seam cell area, with very few 2° branches reaching far from the 1° dendrite (*Figure 2C,G*). These results further supported the notion that the dendritic trapping phenotype in the *kpc-1* mutants was due to the excessive activity of the SAX-7/MNR-1/DMA-1 complex near the seam cells. We also ectopically expressed SAX-7 and MNR-1 in the ALM and PLM neurons in the *sax-7; kpc-1* double mutants (*Figure 2—figure supplement 3*). ALM and PLM neurons have long neurites that extend in the dorsal and ventral sublateral nerve cords and overlap with the locations where PVD 3° branches form and grow. This ectopic expression converted the *sax-7* mutant-like dendritic pattern (*Figure 2B*) to a striking new pattern: the entire PVD dendritic arbor followed the PLM and ALM neurites (*Figure 2—figure supplement 3A*).

Similar to *sax-7; kpc-1*, PVD morphologies of the double mutants between *dma-1* and *kpc-1* also clearly resembled the *dma-1* but not *kpc-1* single mutants, with disrupted organization and complete loss of menorahs but no trapping phenotype (*Figure 2D*, *Figure 2—figure supplement 1D*, *Figure 2—figure supplement 2B,C and H*). The trapping model also predicted that resupplying DMA-1 to *dma-1; kpc-1* double mutants during the outgrowth of 2° branches within the trapping zone would restore the trapping phenotype (*Figure 2D*, middle panel) while expression after the 2° branches reached beyond the trapping zone would not (*Figure 2D*, right panel). To test this, we utilized the heat shock promoter to express DMA-1 at different time points in *dma-1; kpc-1* mutants. Consistent with our hypothesis, providing DMA-1 during early L3 when the 2° branches grew within the trap zone led to robust restoration of the trapping phenotype (*Figure 2E,H*). On the contrary, expressing DMA-1 at later time points, when most of the branches in *dma-1* mutants had already

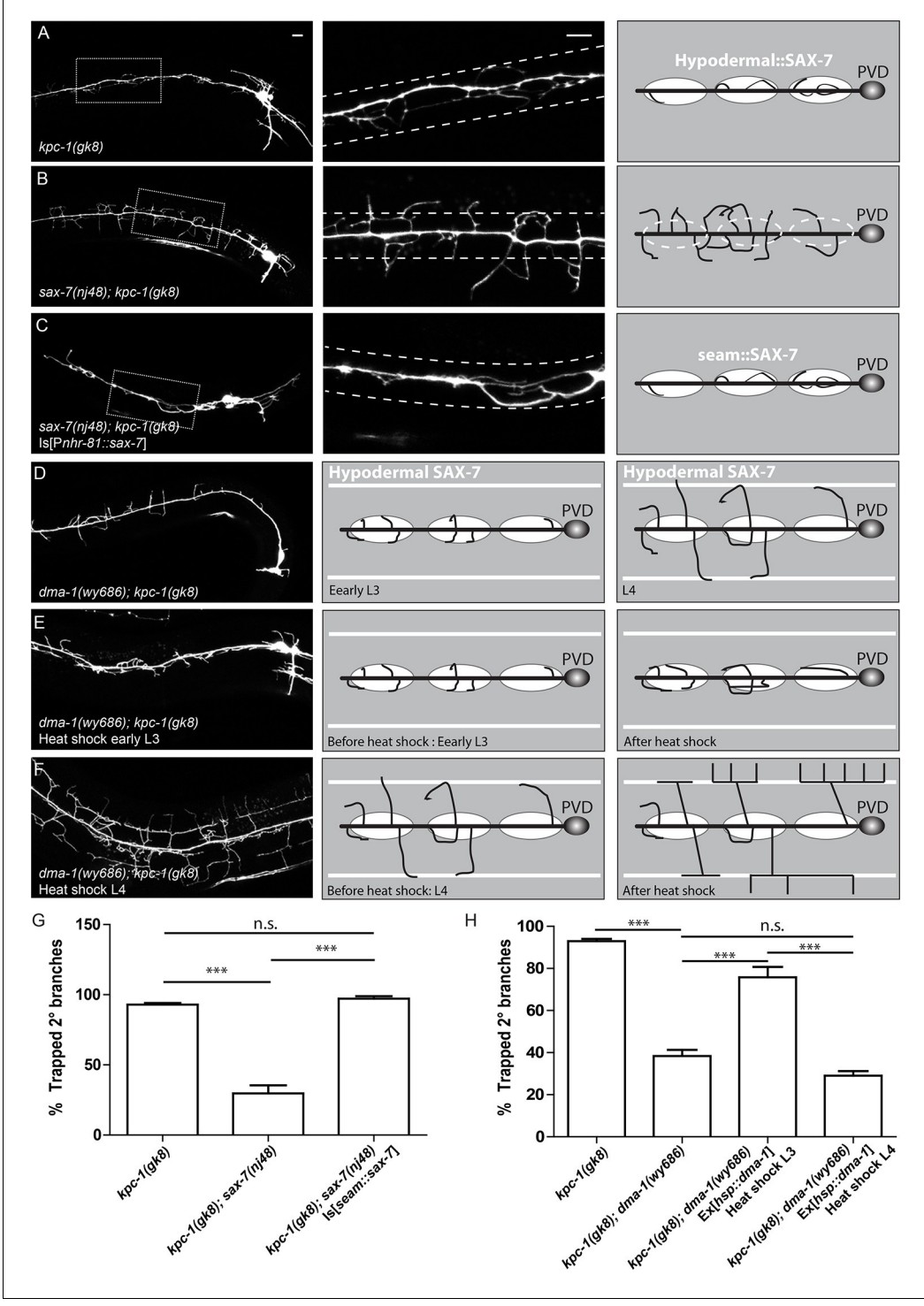

**Figure 2.** The SAX-7/MNR-1/DMA-1 tripartite complex was causal for the trapping phenotype in *kpc-1* mutants. (**A**) Left: Fluorescent image showing PVD morphology of a *kpc-1(gk8)* null mutant. Middle: Zoomed-in view of the boxed area in the left panel. Dotted lines indicate the 'trapping zone' with enriched SAX-7. Almost all 2° dendrites were trapped in this region. Right: Schematic illustration of the phenotype. (**B**) PVD morphologies of *sax-7(nj48); kpc-1(gk8)* double mutants were indistinguishable from *sax-7(nj48)* single mutants. Dendrites could escape from the trap zone. (**C**) Expressing SAX-7 in the seam cells restored the trapping phenotype. (**D**) Left: Fluorescent image showing the PVD morphology of a *dma-1(wy686); kpc-1(gk8)* double mutant. Middle: Schematic illustration showing the initial phase of 2° branch outgrowth during early L3 when the dendrites pass the 'trap zone'. Right: *Figure 2 continued on next page*

*Figure 2 continued*

Later in development, dendrites of the *dma-1; kpc-1* mutants had escaped the trap zone but failed to form menorahs due to lack of DMA-1. (**E**) Expressing DMA-1 during early L3 in *dma-1(wy686); kpc-1(gk8)* double mutants completely restored the trapping phenotype. (**F**) Expressing DMA-1 during L4 or later stages generated a striking rescue of menorahs. Since the dendrites had already escaped from the trap zone, supplying DMA-1 enabled the dendrites to respond to sublateral SAX-7 and MNR-1 signal and form normal 3° and 4° branches at the right place. Scale bar: 10 μm. (G-H) Quantification of the percentage of 2° branches that were trapped around the 1°dendrite. *** is p<0.001, n.s. is p>0.05 by Student's T-test. N=50 for each genotype.

The following figure supplements are available for figure 2:

**Figure supplement 1.** The SAX-7/MNR-1/DMA-1 ligand-receptor complex was causal for the trapping phenotype in *kpc-1* mutants.

**Figure supplement 2.** SAX-7, MNR-1 and DMA-1 were epistatic to KPC-1.

**Figure supplement 3.** PLM and ALM neurons expressing SAX-7-YFP and MNR-1 caused the PVD dendrites of *sax-7; kpc-1* double mutants to follow these neurons.

extended beyond the trapping zone, led to a striking rescue of 3° and 4° branches (*Figure 2F,H*), again demonstrating strongly that dendrites in the *kpc-1* mutants did not lack outgrowth capability and maintained intact SAX-7/MNR-1/DMA-1 signaling. Instead, the *kpc-1* mutant PVD dendrites responded excessively to the guidance cues SAX-7 and MNR-1 near the seam cells, which limited their growth to the 'trap zone'.

## Partial loss of KPC-1 caused defects in self-avoidance of 3° branches and outgrowth of 4° branches

In an independent genetic screen for mutants with PVD dendrite self-avoidance defects, we isolated another allele of *kpc-1, xr58,* which resulted in an amino acid substitution P440S (*Figure 3D*). Unlike a previously reported mutation of the same amino acid (*kpc-1(my24),* P440L) which gave rise to a null phenotype, the PVD neurons of the *kpc-1(xr58)* mutants showed a fully penetrant, complex phenotype (*Figure 3B*) (*Schroeder et al., 2013*). Similar to the complete loss-of-function *kpc-1(gk8)* mutants (*Figure 1B*, *3D*), some 2° dendrites in *kpc-1(xr58)* mutants were trapped near the 1° dendrites (*Figure 3B*, asterisks, *Figure 1C*). The 2° dendrite trapping phenotype was much weaker than that in the *gk8* allele yet significantly different from that in the wild-type animals (*Figure 1C*, *3A–B*). Unlike the PVD dendrites of the *kpc-1* null mutants, which failed to make any intact menorahs, many 2° dendrites in the *kpc-1(xr58)* mutants, especially those in the proximal region near the cell body, could successfully extend away from the 1° dendrite and formed full menorahs with T-shaped 3° branches and many 4° branches on them. However, the 3° branches in this mutant had a striking self-avoidance defect (*Figure 3B*, arrowheads). In wild-type animals, 3° branches showed stringent self-avoidance. Neighboring 3° dendrites almost never overlapped with each other and instead showed gaps in between (*Figure 3A*, arrows). The self-avoidance phenotype of the *xr58* allele was similar to that of two previously reported weak alleles of *kpc-1* (*Salzberg et al., 2014*) whose 3° dendrites remained in the same 2D plane but showed extensive overlap. We quantified the percentage of 'T'-shaped 3° dendrites that made contact with their neighbors and found that around 70% of 3° branches had self-avoidance defects in the *kpc-1(xr58)* mutants, compared to only 2% in wild-type worms (*Figure 5C*). In addition to the self-avoidance defects of the 3° branches, *kpc-1(xr58)* also had a reduced number of 4° branches. Except in the most proximal menorah (closest to the cell body), the 4° dendrites were largely absent from the menorahs (*Figure 3B*, stars, *Figure 5D*).

Several lines of evidence suggested that *kpc-1(xr58)* was a partial loss-of-function allele of *kpc-1*. First, KPC-1 encodes a conserved proprotein convertase and is a homolog of mammalian Furin (*Schroeder et al., 2013*; *Hung et al., 2014*). *kpc-1(xr58)* had a missense point mutation in the catalytic domain of KPC-1, which likely affected its protease function. Injecting a mutated version of *kpc-1* cDNA containing the P440S mutation into the null *kpc-1(gk8)* deletion mutants produced a dendritic phenotype that was milder than *gk8* but indistinguishable from that of *kpc-1(xr58)*, suggesting

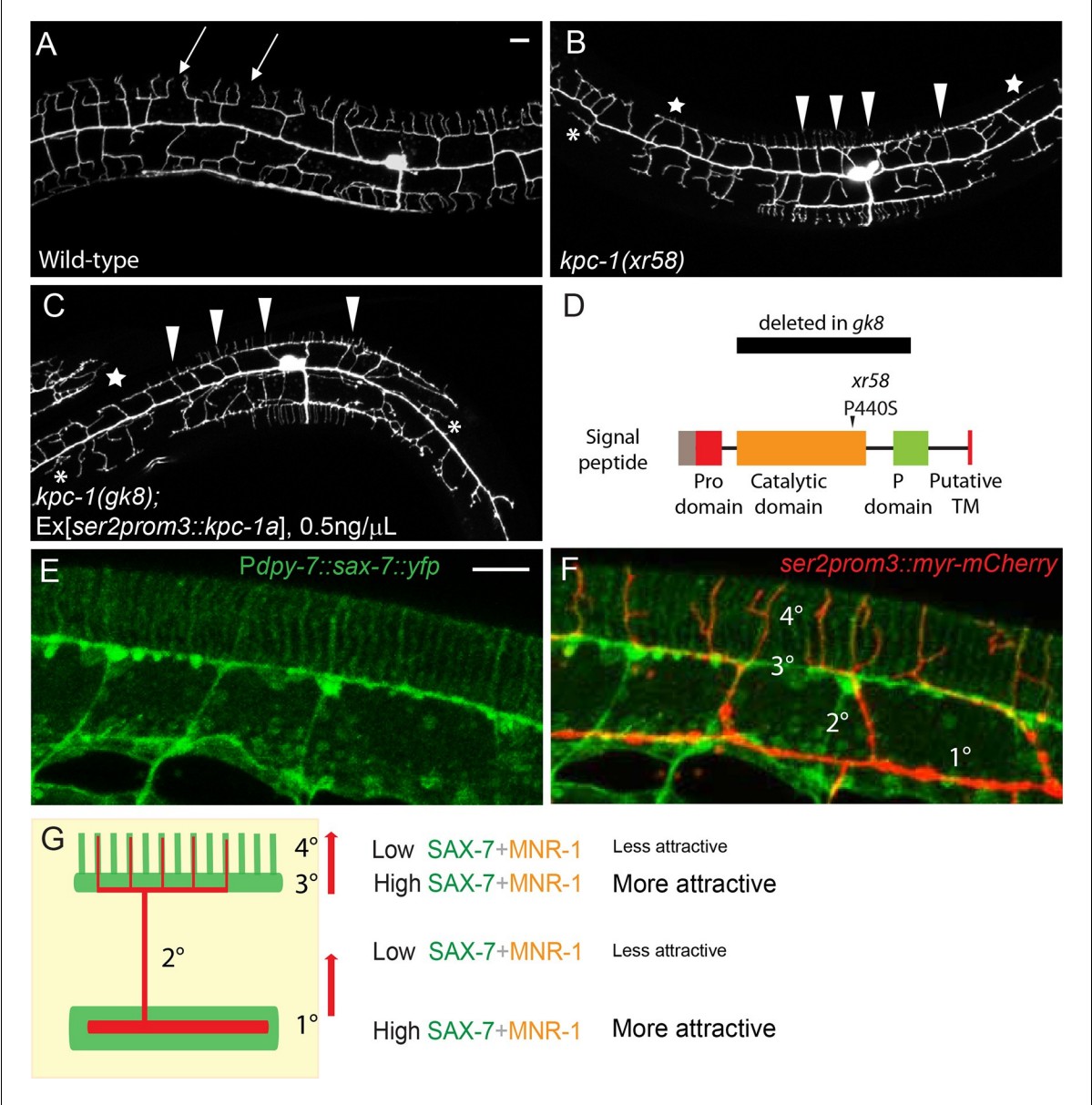

**Figure 3.** Partial loss of KPC-1 caused defects in higher order dendritic branches. (A–C) Fluorescent images showing PVD morphologies of (A) wild-type (B) *kpc-1(xr58)* mutant and (C) *kpc-1(gk8)* mutant animals expressing a low concentration (0.5ng/μL) of full length, functional KPC-1. The *xr58* mutants had severe defects in self-avoidance of 3° branches and reduced number of 4° branches. The phenotype was mimicked by expressing low-level wild-type KPC-1 in the *gk8* null allele of *kpc-1*. Arrows: Gaps between 3° branches in wild-type neurons. Arrowheads: 3° branches that overlapped with their neighbors in mutants. Star: Defective menorah with no 4° branches. Asterisks: Trapped 2° branches. (D) Schematic of the KPC-1 protein showing locations of mutations in *gk8* and *xr58* mutants. (E–F) Fluorescent images of (E) SAX-7 localization in the hypodermal cell, and (F) overlay with a PVD marker in red. SAX-7 was highly enriched in the sublateral lines where 3° branches formed and grew and was also localized to vertical stripes followed by the 4° branches but at much lower concentration. Scale bar: 10 μm. (G) Schematic figure of PVD outgrowth. 2° branches emerging from the 1° dendrites and 4° branches growing away from the 3° branches faced similar challenges to go from regions with higher levels of SAX-7 to places that were less attractive. Compromised function of KPC-1 led to defects in escaping.

The following figure supplement is available for figure 3:

**Figure supplement 1.** Activation of KPC-1 required self-cleavage.

that the *xr58* allele was a hypomorphic allele (data not shown). Second, expressing the wild-type *kpc-1* cDNA at a very low concentration in the *kpc-1(gk8)* null allele resulted in a PVD morphology that was similar to that of the *xr58* mutants (*Figure 3C*, *Figure 3—figure supplement 1D and E*). Third, RNAi against *kpc-1* also gave rise to a similar PVD dendritic phenotype to that of *xr58* (data not shown). Since RNAi was often inefficient in worm neurons, this result further supported the notion that the *kpc-1(xr58)* phenotype represented a partial loss of KPC-1 activity in the PVD neuron.

Despite the seemingly different phenotypes between the null allele and partial loss-of-function allele of *kpc-1*, both phenotypes could be regarded as trapping of the dendritic branches. High levels of SAX-7 were present both around the primary dendrite and along the 3° branch sites (*Liang et al., 2015*), where dendritic branches preferentially grew in *kpc-1* null and partial loss-of-function alleles, respectively (*Figure 3E*). In other words, in wild-type animals, the growth of both the 2° and 4° dendrites required them to move away from the high SAX-7 regions (*Figure 3F,G*). Insufficient KPC-1 function led to failures of dendrites to leave the high SAX-7 intermediate targets.

## Full activation of KPC-1 required self-cleavage and removal of the Pro domain

The full activation of mammalian Furin requires proteolytic cleavage of its N-terminal Pro domain by its own catalytic domain (*Thomas, 2002*). Indeed, when the worm KPC-1 was expressed in cultured *Drosophila* S2 cells, we found two protein products that corresponded to full-length and Pro domain-cleaved KPC-1 based on size (*Figure 3—figure supplement 1F*). With sequence analysis, we identified two adjacent putative cleavage sites in the N-terminus of KPC-1 (*Figure 3—figure supplement 1A*, right panel). We mutated both sites with R to A mutations and found that the resulting mutant protein completely lacked the self-cleavage product when expressed in S2 cells (*Figure 3—figure supplement 1F*), suggesting that these two sites were indeed responsible for the self-cleavage of KPC-1. To assess the physiological function of these cleavage sites, we introduced these two mutations into the endogenous *kpc-1* locus using the CRISPR/Cas9 system (*Paix et al., 2014*). The mutants exhibited a PVD dendritic phenotype indistinguishable from that of the *gk8* null mutants (*Figure 3—figure supplement 1A*), demonstrating the necessity of Pro domain cleavage for the activation of KPC-1. Furthermore, we artificially deleted the Pro domain and expressed this mutant form in the *kpc-1(gk8)* null allele to test if this truncated version of KPC-1 still retained its activity. Consistent with our hypothesis, KPC-1 lacking the Pro domain could fully rescue the *gk8* mutants. Both secondary trapping and menorah outgrowth phenotypes were restored to wild-type level (*Figure 3—figure supplement 1B,D and E*). Since the *xr58* mutation was in the protease domain, we hypothesized that the partial loss of function might be due to inefficient self-cleavage. To test this, we engineered the same P440S mutation into the Pro domain deleted version of KPC-1 and expressed this construct in *kpc-1(xr58)* mutants (*Figure 3—figure supplement 1C*, right panel). Since it was no longer necessary to remove the Pro domain and self-activate, we expected this mutant form to bypass the catalytic defect of *xr58* and restore normal dendritic morphology. KPC-1ΔProP440S fully rescued the 2° branch trapping phenotype (*Figure 3—figure supplement 1C and D*) and partially restored the number of 4° branches (*Figure 3—figure supplement 1C and E*). This was consistent with our model that, similar to mammalian Furin, KPC-1 also needed to cleave its own Pro domain to be fully activated.

## DMA-1 receptor level was increased in *kpc-1* mutants

The dendritic morphology of the *kpc-1(xr58)* mutant was reminiscent of the phenotype caused by overexpression of DMA-1 in PVD (*Figure 5A*) (*Liu and Shen, 2012*): First, neurons overexpressing DMA-1 had severe self-avoidance defects. Their 3° dendrites often failed to avoid each other and formed a continuous fascicle that covered the entire sublateral line (*Figure 5A*, arrowheads, *Figure 5C*). In addition, many 2° dendrites were trapped around the 1° dendrite (*Figure 5A*, asterisks, *Figure 1C*), and the total number of 4° branches was decreased (*Figure 5A*, star, *Figure 5D*). All three aspects of the DMA-1 overexpression phenotype were similar to those of the *kpc-1(xr58)* mutants, suggesting that having too much DMA-1 led to similar consequences as insufficient KPC-1 activity. Since both KPC-1 and DMA-1 functioned autonomously in the PVD neuron, we hypothesized that KPC-1 might affect the PVD dendrites through regulating DMA-1.

We utilized several approaches to directly examine DMA-1 in *kpc-1* mutants. First, we visualized the DMA-1 protein with an integrated reporter expressing *gfp*-tagged *dma-1* genomic DNA driven by the PVD-specific *ser2prom3* promoter. This transgenic reporter allowed us to monitor the amount of DMA-1 receptors at precise locations in the entire PVD dendritic arbor. In wild-type animals, diffuse DMA-1-GFP fluorescence was localized to all dendritic compartments including the higher order branches (*Figure 4A*). Discrete GFP puncta could also be found in the cell body and 1° dendrites, which were likely secretory or endocytic vesicles that carry DMA-1 (*Figure 4A*, right panel). In wild-type neurons, DMA-1-GFP intensity in the discrete puncta was significantly higher than the diffuse staining in the dendrites (*Figure 4A*). In the *kpc-1(gk8)* mutants, we observed a striking increase in the diffuse DMA-1-GFP intensity in the entire dendritic arbor (*Figure 4B*). Quantification of the brightness of diffuse DMA-1-GFP on the dendrite, which was likely membrane-localized DMA-1-GFP, showed a dramatic increase in the *kpc-1(gk8)* allele compared with wild-type (*Figure 4C*). A similar but less dramatic increase in DMA-1-GFP intensity was observed for the weak allele *kpc-1 (xr58)* as well (*Figure 4C*). Such up-regulation could be completely rescued by PVD autonomous expression of KPC-1 (*Figure 4C*). To quantify the total DMA-1-GFP protein level, we performed Western blot analysis using an antibody against GFP and confirmed that there was indeed an overall increase in DMA-1 protein level in the *kpc-1* mutants (*Figure 4D*). Furthermore, we monitored endogenous DMA-1 by engineering a YFP into the *dma-1* genomic locus using CRISPR (*Paix et al., 2014*). Despite dim fluorescence intensity, we were able to visualize the endogenous expression and observed a DMA-1 localization pattern similar to what we saw using high copy transgenes, with both membrane and vesicular distribution of the protein (data not shown). When we crossed the *dma-1:: yfp* knockin strain into *kpc-1(gk8)*, we again observed dramatic increase in DMA-1-YFP intensity in the double mutants (*Figure 4—figure supplement 1D*). Similarly, a 2xFLAG tag was inserted in frame into the endogenous locus of *dma-1* using CRISPR (*Paix et al., 2014*), and we again measured significant up-regulation of DMA-1 protein level in *kpc-1* mutants (*Figure 4D,E*). As a control, we examined the fluorescent intensity and localization of HPO-30, another transmembrane protein important for PVD development (*Smith et al., 2013*), and observed no significant change in its dendritic levels in the *kpc-1(gk8)* mutants despite severe morphology defects (*Figure 4—figure supplement 1A–C*). We also performed qPCR and found that there was no difference in *dma-1* transcript level between wild-type and *kpc-1(gk8)* mutants (*Figure 4—figure supplement 1E*). These results further supported the notion that KPC-1 specifically down-regulated the DMA-1 protein level without affecting transcription.

We noticed an increase in branching in the proximal region when we overexpressed DMA-1 in the *kpc-1* mutants (*Figure 4B*, *Figure 4—figure supplement 2A*). Although most dendrites were still trapped around the 1° neurite (*Figure 4—figure supplement 2A*, right panel), some 2° dendrites close to the soma extended away to form 3° branches. We suspected that the very high level of DMA-1 on the dendritic membrane caused by both the lack of KPC-1 and the DMA-1 overexpression transgene might saturate the SAX-7/MNR-1 ligands in the region, allowing the dendrites to escape. To test this hypothesis, we further overexpressed SAX-7 and MNR-1 in the hypodermal cells in this genetic background, and indeed saw that the dendrites were trapped near the seam cells again (*Figure 4—figure supplement 2B*).

Genetic interactions between *kpc-1* and *dma-1* were also in support of this model. First, overexpressing DMA-1 in the *kpc-1(xr58)* allele further enhanced the dendritic phenotypes (*Figure 5B*). The 3° branches in these animals showed an even more severe self-avoidance defect than the *kpc-1 (xr58)* worms, and the number of 4° branches was further reduced (*Figure 5C,D*). Second, we found that overexpression of a truncated version of DMA-1 lacking its entire cytosolic domain generated a dramatic trapping phenotype that mimicked that of the *kpc-1(gk8)* null mutant (*Figure 5E*). In these animals, 2° dendrites were short and trapped around the 1° dendrite. We suspected that the cytosolic domain of DMA-1 may contain signaling motifs that were important for its endocytosis, so overexpressing a DMA-1 construct which lacked these motifs led to higher level of DMA-1 on the dendritic membrane than overexpression of the full length DMA-1 did, thereby produced a more severe trapping phenotype. The same construct, when expressed at a much lower level could partially rescue the *dma-1* mutant phenotype by allowing the formation of 3° and a small number of 4° branches (*Figure 5—figure supplement 1D*). We also generated a similar mutant allele of *dma-1* lacking its cytosolic domain using the CRISPR/Cas9 system (*Friedland et al., 2013*) and observed a similar phenotype. 2° and 3° branches of these mutants developed normally, suggesting that this

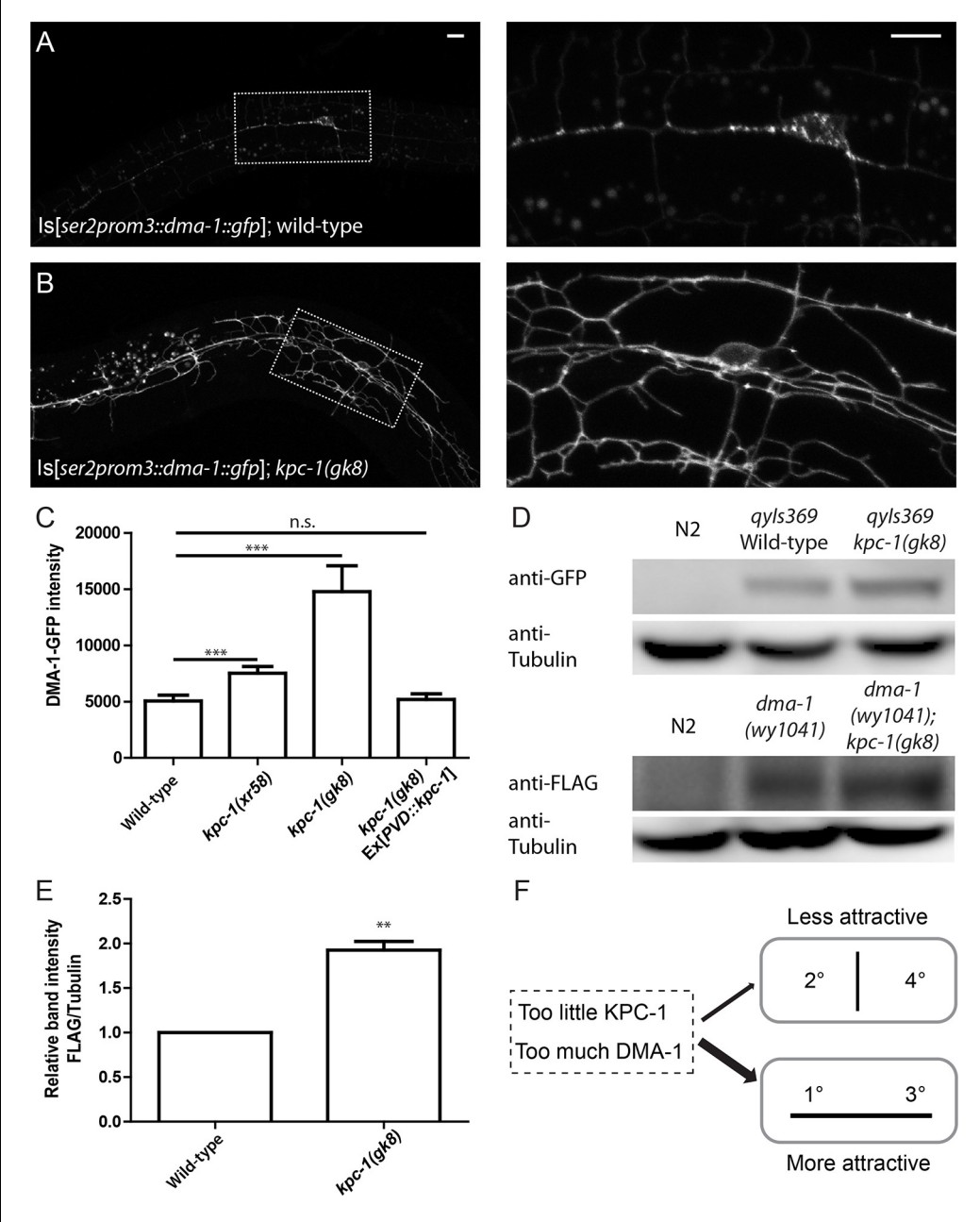

**Figure 4.** The level of the DMA-1 receptor was increased in *kpc-1* mutants. (**A–B**) Fluorescent images of DMA-1::GFP in PVD neurons in (**A**) wild-type and (**B**) *kpc-1(gk8)* mutant animals. DMA-1 showed diffuse staining in the entire dendritic arbor but was more enriched in vesicles in the cell body and 1° dendrites in wild-type PVD. Diffuse membrane localization of DMA-1::GFP was significantly increased and vesicle-like puncta were reduced in the *kpc-1(gk8)* mutant. The images on the right are zoomed-in views of the regions indicated by the dashed boxes. Scale bar: 10 μm. (**C**) Quantification of fluorescent intensity of diffuse DMA-1::GFP on the 2° dendrites. *** is p<0.001 by Student's T-test. N=50 for each genotype. (**D**) Upper panels: Western blot against GFP in wild-type worms without transgene, wild-type worms expressing DMA-1-GFP and *kpc-1(gk8)* mutant worms expressing DMA-1-GFP. Lower panels: Western blot against FLAG in wild-type animals, *dma-1(1041)* mutants with 2xFLAG inserted into the *dma-1* cytosolic domain of the endogenous genomic locus using CRISPR/Cas9, and *dma-1(1041); kpc-1(gk8)* double mutants. (**E**) Quantification of relative band intensity normalized to α-tubulin. ** is p<0.01 by Student's T-test. N=4 (**F**) Schematic figure of the proposed model in which loss of KPC-1 caused increased membrane DMA-1, leading to defects in escaping from the high levels of ligands around the 1° and 3° dendrites.

*Figure 4 continued on next page*

*Figure 4 continued*

The following figure supplements are available for figure 4:

**Figure supplement 1.** KPC-1 caused specific down-regulation of DMA-1 receptor.

**Figure supplement 2.** (A) Overexpressing DMA-1 caused more dendrites to escape from the trap zone.

truncated protein was partially functional (*Figure 5—figure supplement 1A,C*). Interestingly, removing KPC-1 in this genetic background resulted in a severe trapping phenotype, indicating that KPC-1 did not regulate DMA-1 through its cytosolic domain and that the extracellular domain of DMA-1 was sufficient to respond to SAX-7/MNR-1 and to generate trapping (*Figure 5—figure supplement 1B*). The striking similarity between the *kpc-1* null and the strain overexpressing the truncated DMA-1 shown in *Figure 5E*, together with the observation that the loss of DMA-1 suppressed the trapping phenotype of the *kpc-1(gk8)* mutant allele, argued strongly that the overabundance of the DMA-1 receptor was responsible for the phenotypes in the loss-of-function alleles of *kpc-1* (*Figure 4F*).

## KPC-1 reduced the binding between DMA-1 and SAX-7/MNR-1 *in vitro*

To test whether KPC-1 directly affected the interaction between DMA-1 and its ligands SAX-7 and MNR-1, we co-expressed KPC-1 and DMA-1 in *Drosophila* S2 cells and asked whether these cells could still form aggregates with cells expressing SAX-7/MNR-1. As we have reported previously, when DMA-1-RFP expressing cells were mixed with cells co-expressing SAX-7-GFP and MNR-1-GFP, the red and green cells formed clusters that indicated direct interaction between the DMA-1 receptor and the SAX-7/MNR-1 complex *in trans* (*Figure 6A*, arrows, *Figure 6C*) (*Dong et al., 2013*). Co-transfection of KPC-1 with DMA-1 in the same cells completely blocked cell aggregation (*Figure 6B, C*). We then directly examined the amount of DMA-1 in the cells by probing for Myc-tagged DMA-1 with an anti-Myc antibody on Western blots, and found that, consistent with what we have observed in worms, the DMA-1 level was diminished in the cells co-expressing KPC-1 (*Figure 6D,E*). Another type I transmembrane protein mCD8 was co-transfected in the same cell cultures and was not affected, showing that the effect of KPC-1 on DMA-1 was specific. This result showed that KPC-1 down-regulated DMA-1 not only in worm neurons but also in *Drosophila* S2 cells.

Since DMA-1 functioned as the membrane receptor for dendritic branching, we examined the membrane localization of DMA-1 and how it was modified by KPC-1 in adhesive S2R+ cells. DMA-1-RFP alone localized to the plasma membrane, which could be seen particularly clearly on the filopodial and lamellipodial structures (*Figure 6—figure supplement 1A*). However, when KPC-1-CFP was co-transfected with DMA-1-RFP, membrane-localized DMA-1-RFP was diminished (*Figure 6—figure supplement 1B*). In contrast, neither the level nor the localization of mCD8 was affected by KPC-1.

## KPC-1 targeted DMA-1 to endosomes via direct interaction with its extracellular domain

Receptor down-regulation is often achieved by targeting the transmembrane proteins to endosomal pathways either from the trans-Golgi network (TGN) or via endocytosis from the plasma membrane (*Katzmann et al., 2002*; *Maxfield and McGraw, 2004*). We first asked whether DMA-1 was targeted to late endosomes and lysosomes in wild-type worms. An extrachromosomal array expressing mCherry-RAB-7 marker which labeled the late endosomes/lysosomes (*Poteryaev et al., 2010*) was co-expressed with the integrated DMA-1-GFP line to visualize DMA-1 and the late endosomes simultaneously. Indeed, the intracellular DMA-1-GFP puncta showed a high degree of co-localization with mCherry-RAB-7 in wild-type PVD neurons (*Figure 7A*). Around 80% of RAB-7 vesicles co-localized with DMA-1-GFP (*Figure 7E*). This result indicated that DMA-1 was indeed targeted to late endosomes in wild-type animals. In contrast, in *kpc-1(gk8)* mutant animals, concomitant with the increased DMA-1-GFP level on the dendritic membrane, there was reduced co-localization between DMA-1-GFP and RAB-7 vesicles (*Figure 7B,E*). We hypothesized that surface DMA-1 could be down-regulated via two distinct mechanisms: One that required the cytosolic domain of DMA-1 but was independent of KPC-1, and the KPC-1 pathway. The fact that overexpression of cytosolic truncated DMA-1 produced a much stronger gain-of-function phenotype than full length DMA-1 (*Figure 5E*) argued that cytosolic region-mediated endocytosis of DMA-1 might help to down-

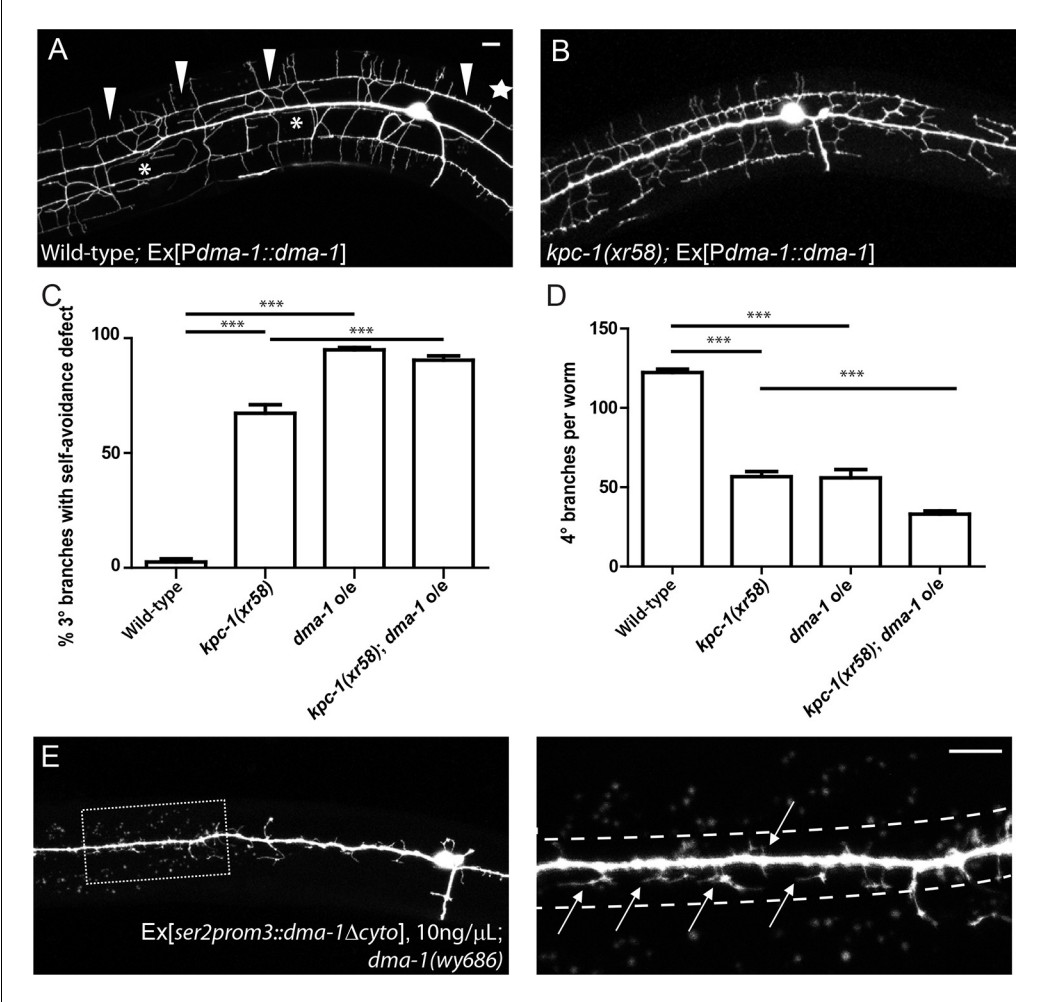

**Figure 5.** Overexpression of DMA-1 generated *kpc-1* mutant-like phenotypes. (**A**) Overexpression of DMA-1 in PVD neurons caused similar defects to those of the *kpc-1(xr58)* mutants shown in *Figure 3B*. Arrowheads: 3° branches that overlapped with their neighbors in mutants. Star: Defective menorah with no 4° branches. Asterisks: Trapped 2° branches. (**B**) Overexpressing DMA-1 in *kpc-1(xr58)* enhanced the 3° self-avoidance and 4° outgrowth phenotypes. (**C**) Quantification of the percentage of 3° branches that made contact with their neighbors. (**D**) Quantification of the total number of 4° branches per animal. *** is p<0.001 by Student's T-test. N=50 for each genotype. (**E**) Overexpressing truncated DMA-1 without its cytosolic domain produced dramatic trapping phenotype. Arrows: Trapped dendrites. Dotted lines indicated the trap zone. Scale bar: 10 μm.

The following figure supplement is available for figure 5:

**Figure supplement 1.** Regulation of DMA-1 by KPC-1 did not require the cytosolic domain of DMA-1.

regulate its level on the plasma membrane. Indeed, DMA-1ΔcytoGFP exhibited much stronger membrane localization when expressed at comparable levels as the full length protein (*Figure 7C*). We then quantified the co-localization between mCherry-RAB-7 and DMA-1Δcyto-GFP and observed a reduction in the percentage of RAB-7 vesicles containing GFP (*Figure 7C and E*). Furthermore, when we crossed the integrated transgene expressing DMA-1Δcyto-GFP into *kpc-1(gk8)* mutants, the GFP signal appeared almost exclusively on the plasma membrane (*Figure 7D*). Very few vesicles were seen throughout the dendritic arbor and only 10% of late endosomes labeled by mCherry-RAB-7 contained DMA-1Δcyto-GFP (*Figure 7D and E*). Since both DMA-1 and KPC-1 were predicted to be type I transmembrane proteins, and since the down-regulation of DMA-1 by KPC-1 did not require the cytosolic domain, we reasoned that KPC-1 might target DMA-1 for degradation through directly interacting with its extracellular domain. To test this hypothesis, DMA-1 and KPC-1 ectodomains were expressed using baculoviruses in High Five cells and purified. Direct binding

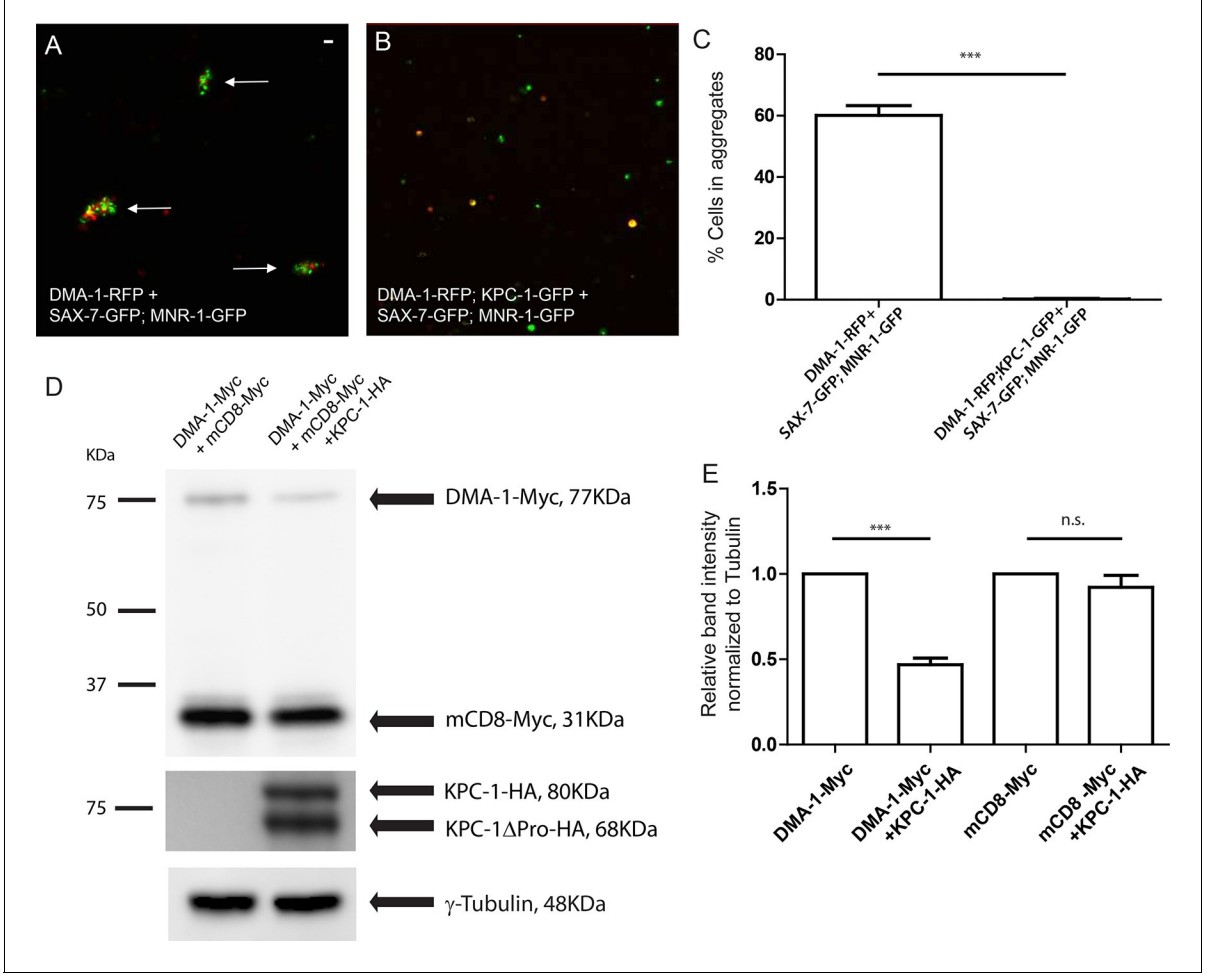

**Figure 6.** KPC-1 interrupted the interaction between DMA-1 and SAX-7/MNR-1 by down-regulating membrane DMA-1. (A) *Drosophila* S2 cells co-expressing SAX-7-GFP and MNR-1-GFP formed aggregates with cells expressing DMA-1-RFP alone. (B) Cell aggregation failed when KPC-1-GFP is co-transfected with DMA-1-RFP. (C) Quantification of percentages of fluorescent cells in aggregates after 3 hr. *** is p<0.001 by Student's T-test. The experiment was repeated three times for quantification. (D) Immunoblot showing that the amount of DMA-1 was significantly reduced when co-transfected with KPC-1 while that of another co-transfected type I transmembrane protein, mCD8, was not affected. (E) Quantification of band intensity on the Western blots. *** is p<0.001 and n.s. is p>0.05 by Student's T-test. Each experiment was repeated three times for quantification.

The following figure supplement is available for figure 6:

**Figure supplement 1.** KPC-1 prevented DMA-1 from localizing to the plasma membrane.

between these proteins was detected with an apparent dissociation constant of 33 μM using biolayer interferometry (*Figure 7F*, blue circles). No binding was detected between DMA-1 and the negative control, the ectodomain of human GPR56 (*Figure 7F*, open rectangles).

We made several attempts to visualize KPC-1 in the PVD neuron. KPC-1 tagged on both its N- and C-termini produced very weak fluorescent signal. We thus turned to the S2R+ cells, in which a similar down-regulation effect of KPC-1 on DMA-1 had been observed (*Figure 6—figure supplement 1A,B*). We co-expressed KPC-1-CFP and Venus-RAB-7 in these cells and observed strong co-localization between KPC-1 and RAB-7 (*Figure 7G–I*, arrows). Together, these data were consistent with a model in which KPC-1 bound directly to the ectodomain of DMA-1 and targeted the receptor to late endosomes and lysosomes for degradation (*Figure 7J*). In *kpc-1* mutants, the level of DMA-1 was increased on the dendritic plasma membrane, leading to excessive responsiveness to the guidance signals SAX-7 and MNR-1. Consequently, PVD dendrites became trapped at intermediate targets with enriched SAX-7 and failed to move away.

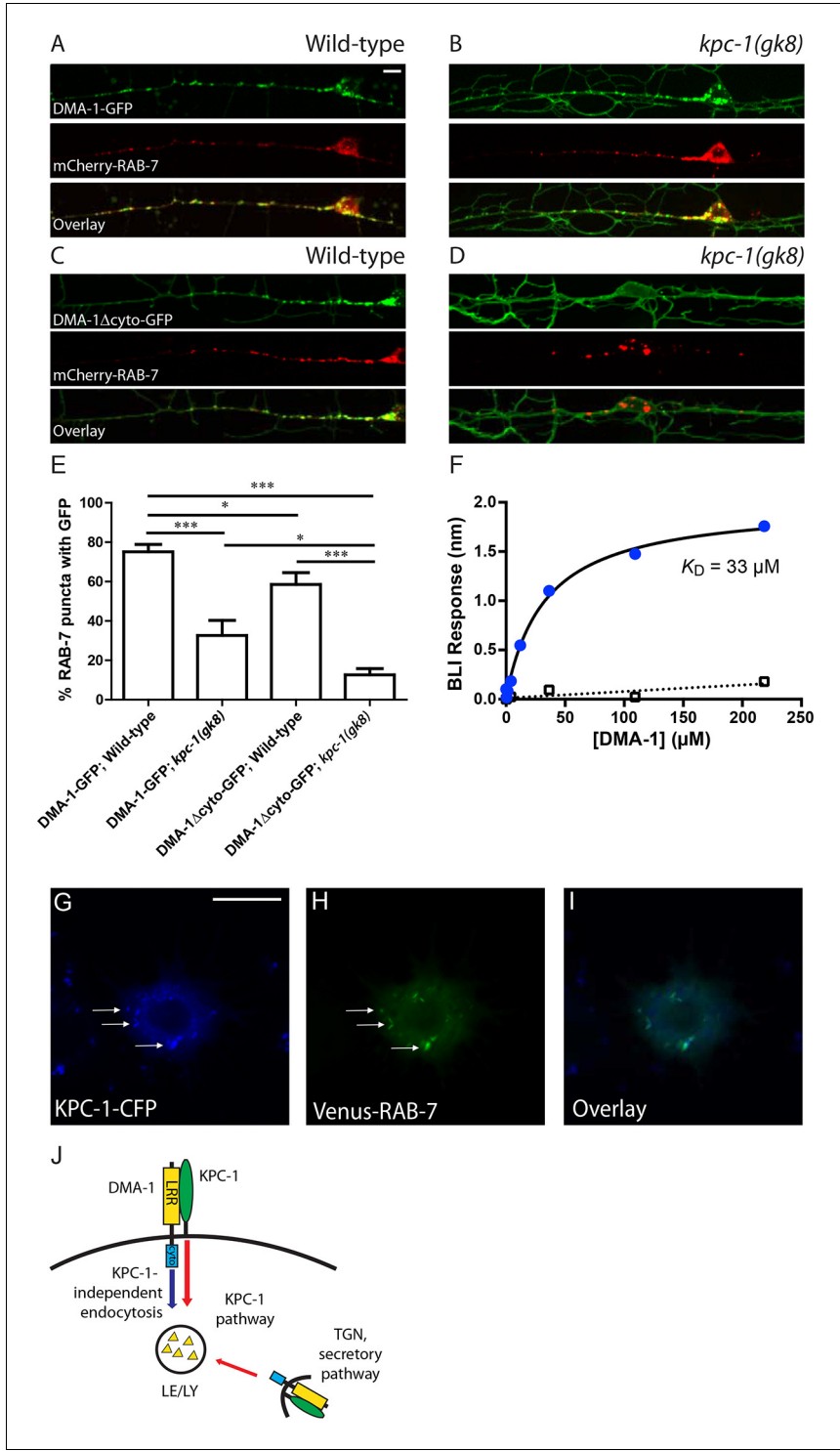

**Figure 7.** KPC-1 targeted DMA-1 to endocytic vesicles through direct interaction with its ectodomain. (**A–B**) Fluorescent images of DMA-1-GFP (upper panels), mCherry-RAB-7 (middle panels) and overlay (lower panels) in PVD neurons of (**A**) wild-type and (**B**) *kpc-1(gk8)* mutant animals. Many DMA-1-GFP puncta co-localized with late endosomes/lysosomes labeled by mCherry-RAB-7 in wild-type PVD while the co-localization was reduced in the *kpc-1(gk8)* mutants. *kpc-1* mutants showed enhanced DMA-1-GFP fluorescence on the membrane of dendritic branches but less in vesicles. (**C–D**) Fluorescent images of DMA-1Δcyto-GFP (upper panels), mCherry-RAB-7 (middle panels) and overlay (lower panels) in PVD neurons of (**C**) wild-type and (**D**) *kpc-1(gk8)* mutants. DMA-1 lacking its entire cytosolic domain showed brighter signal on the plasma membrane but still localized to endocytic

*Figure 7 continued on next page*

*Figure 7 continued*

vesicles. However, in the *kpc-1(gk8)* mutants, DMA-1Δcyto-GFP was almost exclusively on the plasma membrane but was absent from RAB-7-positive versicles. Scale bar: 10 μm. (**E**) Quantification of the percentages of mCherry-RAB-7 vesicles that showed DMA-1-GFP or DMA-1Δcyto-GFP fluorescence. *** is p<0.001, * is p<0.05 by Student's T-test. N=20 for each genotype. (**F**) DMA-1 binding on KPC-1-immoilized surface using biolayer interferometry. Blue circles represent DMA-1 binding responses on KPC-1, with the black curve as the fit to a Langmuir isotherm model. Open rectangles show DMA-1 binding on a negative-control surface with the SA tip decorated with biotynlated ectodomain of human GPR56. (**G-I**) Fluorescent images showing the localization of (**G**) KPC-1-CFP, (**H**) Venus-RAB-7 and (**I**) overlay in S2R+ cells. Arrow: KPC-1-CFP puncta that co-localized with Venus-RAB-7. Scale bar: 10 μm. (**J**) Schematic illustration of the model. Plasma membrane DMA-1 is down-regulated via two synergistic mechanisms: KPC-1 binds to the ectodomain of DMA-1 and targets it to endosomes, while other endocytic pathways signal through the cytosolic domain of DMA-1. LRR: Leucine rich repeats domain, cyto: cytosolic domain, LE/LY: Late endosomes/lysosomes

## Discussion

### Receptor level dictates the neurites' response to adhesive guidance cues

Receptor-ligand interaction is an important theme in neural development. It has been shown by the classic studies on axon guidance that the same ligand can trigger a variety of responses in different neuronal types depending on the membrane receptor contents of their growth cones. Netrin functions as a chemoattractant when binding to the UNC-40/DCC receptor but repels axons containing an UNC-5 and UNC-40/DCC receptor complex (*Hedgecock et al., 1990*; *Tessier-Lavigne and Goodman, 1996*; *Hong et al., 1999*; *Powell et al., 2008*). In regions of the nervous system that are wired in a topographic fashion such as the superior colliculus in the visual system, the relative level of Eph receptors dictates where in the Ephrin gradient an axon stops (*Cheng et al., 1995*; *Wilkinson, 2001*; *Mann et al., 2002*). Even the same neuron can respond to a ligand differently at different decision points during development. In *Drosophila*, Comm functions in pre-crossing commissural axons to keep the Robo receptor level low by sorting the receptors into endosomes, whereas Comm is turned off after the axon has crossed the midline and the Robo receptor is restored on the surface of growth cones to gain responsiveness to the midline repellent Slit and prevent axons from recrossing (*Keleman et al., 2002*; *Keleman et al., 2005*; *Dickson and Gilestro, 2006*). Similarly, commissural axons in the mammalian spinal cord change their responses to Slit by altering their expression of the Robo3 splicing isoforms. (*Sabatier et al., 2004*; *Chen et al., 2008*). Hence, the temporal and spatial regulation of receptors is critical to achieving precise neural development.

We present a new case in neurite morphogenesis in which dynamic regulation of the guidance receptor is required for proper development. Like axons, dendrites face the challenge of ignoring and escaping from intermediate targets with high affinity guidance cues after reaching them. We show that KPC-1/Furin is responsible for down-regulation of the DMA-1 receptor to allow dendrites to move away from an area with enriched SAX-7/MNR-1 ligands.

### KPC-1 functions in the PVD neuron to down-regulate the DMA-1 receptor

KPC-1 is a worm proprotein convertase known to process and activate proteins and peptides by proteolytic cleavage at a conserved consensus sequence (*Thomas, 2002*). For example, KPC-1 was shown to be a major convertase to cleave and activate insulin peptides in neurons for the activation of the dauer formation pathway (*Hung et al., 2014*). Our results, like previous studies, indicate that KPC-1 functions in multi-dendritic neurons to assist dendritic morphogenesis and is in the same genetic pathway as the SAX-7/MNR-1/DMA-1 signaling complex (*Schroeder et al., 2013*; *Salzberg et al., 2014*). However, our results provide strong genetic and cell biological evidence that the morphological defects of the PVD dendrites in the *kpc-1* mutants are due to enhanced, rather than decreased, response to the SAX-7/MNR-1 ligands.

First, we showed a causal relationship between the presence of the SAX-7/MNR-1/DMA-1 tripartite complex and the dendritic trapping phenotype of the *kpc-1* mutants. Dendrites of the *kpc-1* mutants were trapped in regions with high levels of SAX-7. In the null allele of *kpc-1*, almost all 2°

dendrites were unable to extend away from the hypodermal-seam cell junction region where SAX-7 was highly enriched. Removal of SAX-7 or other components of the tripartite complex completely suppressed the trapping phenotype in the *kpc-1* null mutants. Dendritic morphologies of the *kpc-1; sax-7/mnr-1/dma-1* double mutants were indistinguishable from the *sax-7, mnr-1* or *dma-1* single mutants but were drastically different from the *kpc-1* null mutants. Thus, the outgrowth defect of *kpc-1* mutants was due to the trapping of dendrites caused by excessive response to SAX-7 and MNR-1. To further confirm this model, we used ectopic expression of SAX-7 to manipulate ligand distribution and tested if that was sufficient to cause a predictable trapping pattern. Indeed, we could trap the *kpc-1* mutant dendrites ectopically by expressing SAX-7 and MNR-1 in seam cells or PLM and ALM neurons (*Figure 2*, *Figure 2—figure supplement 3*). Also, when we expressed the DMA-1 receptor at a late developmental stage in *dma-1; kpc-1* double mutants when dendrites had 'escaped' from the trap zone, menorah structures could be largely rescued, showing that PVD's ability to extend dendrites and respond to the ligand complex was intact in *kpc-1* mutants.

Second, we showed that the DMA-1 receptor was down-regulated by KPC-1 both endogenously in PVD neurons and in heterologous cell systems (*Figure 4*, *Figure 6*). Overexpression of a cytoplasmic domain deleted version of DMA-1 mimicked the *kpc-1(gk8)* phenotype (*Figure 5E*), showing that the dendritic defects of *kpc-1* mutants could be attributed to excessive amounts of DMA-1 on the membrane. In the weak *kpc-1(xr58)* allele, overexpression of DMA-1 enhanced the 3° self-avoidance and the 4° outgrowth phenotypes, which further supported the notion that the phenotypes arose as a result of elevation in DMA-1 level.

How does KPC-1 down-regulate DMA-1? Both proteins are predicted to be type I transmembrane proteins, with KPC-1 having a C-terminal transmembrane domain but no cytosolic domain. The topology of these proteins suggested that KPC-1 could cleave DMA-1's ectodomain at specific sites and inactivate DMA-1. We could not, however, identify any consensus Furin cleavage sites in the DMA-1 protein sequence. Likewise, we did not observe any truncated product of DMA-1 in the presence of KPC-1 (*Figure 4—figure supplement 1F*). Instead, we observed a reduction in the total amount of DMA-1 when KPC-1 was co-expressed in systems that we examined. These results suggested that KPC-1 did not cleave DMA-1 but rather down-regulated DMA-1 via a different mechanism. We also found that, consistent with the proteins' topologies, the regulation of DMA-1 by KPC-1 did not require its cytosolic domain, since double mutants between a *dma-1* allele lacking its cytosolic domain *dma-1(wy908)* and *kpc-1(gk8)* still showed strong trapping phenotype (*Figure 5—figure supplement 1B*). We therefore hypothesized that KPC-1 functioned similarly to another member of the mammalian convertase family, PCSK9, which down-regulated the LDL receptor through directly binding to its extracellular EGF domain and targeting the receptor to the endocytic pathway for degradation (*Lagace et al., 2006*; *Zhang et al., 2007*; *Poirier and Mayer, 2013*). Consistent with this model, we detected direct interaction between the ectodomains of DMA-1 and KPC-1 and observed that KPC-1 localized to the late endosomes and lysosomes when expressed in S2R+ cells.

In summary, we show that KPC-1 down-regulates the DMA-1 receptor through direct interaction with its extracellular domain in PVD neurons. This function is necessary to allow developing dendrites to move away from the intermediate targets where the guidance cues SAX-7 and MNR-1 are highly enriched before reaching their final destinations. These results demonstrate that precise regulation of receptor levels is critical for dendrite branching and patterning.

## Materials and methods

### Strains and plasmids

N2 Bristol was used as the wild-type strain. Worms were raised on OP50 *Escherichia coli (E. coli)*-seeded nematode growth medium (NGM) plates at 20°C or room temperature, following standard protocol (*Brenner, 1974*). All transgenes and plasmids are listed in *Supplementary file 1*-Tables S1 and S2.

### Generation of the *dma-1(908)*, *dma-1(wy1041)* and *kpc-1(wy1060)* alleles with CRISPR/Cas9

To generate new *dma-1* alleles by CRIPSR/Cas9-mediated genome editing, P*eft-3::cas9* (50 ng/μL), p*U6::dma-1* sgRNA (50 ng/μL), P*unc-122::rfp* (50 ng/μL) and P*myo-3::mCherry* (5 ng/μL) were

injected into *wyIs592 (ser2prom3:: myr::gfp)* worms. F1 worms with abnormal PVD dendrite morphologies were identified through fluorescence microscopy and rescued. 8 of 24 F1 worms carried germ line mutations (point mutations, small insertions/deletions) in *dma-1*, which were revealed by sequencing. *wy908* contained an 8 base pairs deletion that caused a frame shift leading to premature stop codons in the cytosolic domain of *dma-1*.

A similar approach was used to insert 2xFLAG into the cytosolic domain of *dma-1* using CRISPR/Cas-9 combined with homologous repair. A single-stranded repair template carrying 2xFLAG and 50 base pairs of homology sequences on each end was co-injected at 50ng/µL and successful insertion was confirmed by PCR genotyping and sequencing. The FLAG tag was inserted between the third and fourth amino acids after the transmembrane domain, and we confirmed with fluorescent microscopy that these mutants had normal PVD morphology and thus the edited *dma-1* allele was still fully functional.

The *kpc-1(wy1060)* allele was generated using a 113 base pairs single-stranded repair template. The 136th and the 143th Arginines were mutated to Alanines. The entire coding region of *kpc-1* was sequenced to confirm precise homologous recombination and no other mutations were identified. Sequences of all sgRNAs and repair oligos are listed in *Supplementary file 1*-Table S3.

## Fluorescence microscopy and confocal imaging

Images of DMA-1-GFP and HPO-30-GFP were acquired in live animals using a Zeiss Axio Observer Z1 microscope equipped with a Plan-Apochromat 63X/1.4NA objective, Yokagawa spinning disk head, 488 nm and 561 nm diode lasers, and a Hamamatsu ImagEm EMCCD camera driven by MetaMorph (Molecular Devices, Sunnyvale, CA). Other fluorescent images were captured using a Plan-Apochromat 40X/1.3NA objective on a Zeiss LSM710 confocal microscope (Carl Zeiss, Germany). Worms were immobilized on 2% agarose pads using a mixture of 225 mM 2,3-butanedione monoxime and 2.5 mM levamisole (Sigma-Aldrich, St. Louis, MO). Z-stacks were collected and maximum intensity projections were used for additional analysis.

## S2 aggregation and S2R+ cell transfection

*Drosophila* S2 and S2R+ cells were obtained from the *Drosophila* Genomics Resource Center and cultured in Schneider's insect medium (Sigma) according to the manufacturer's description and transfected using Effectene (Qiagen, Valencia, CA). S2 cell aggregation assays were performed as previously described (*Zorio et al., 1994*). All plasmids used for transfection are listed in *Supplementary file 1*-Table S2. For the aggregation assay, S2 cells were transfected with P*actin:: sax-7::gfp* +P*actin::mnr-1::gfp*, P*actin::dma-1::rfp* or P*actin::dma-1::rfp*+ P*actin::kpc-1::gfp*. 3 days after transfection, cells were washed with 5 mL phosphate buffered saline (PBS) and resuspended in S2 medium at $10^6$ cell/mL. 500 µL of green cells were mixed with 500 µL red cells and rotated at 30 rpm for 3 hours at room temperature. 3 µL of each mixture was immediately spotted on glass slides for imaging and quantification.

Similarly, S2R+ cells were cultured in 4-well chamber slides (Fisher Scientific, Waltham, MA) and transfected with P*actin::dma-1::rfp*, P*actin::mCD8::Venus*, P*actin::kpc-1::cfp* and P*actin::Venus::rab-7.* The cells were washed with PBS and imaged on a Zeiss LSM710 confocal microscope (Carl Zeiss) 2 days after transfection.

## Protein expression in S2 cells and Western blotting

For *Figure 3—figure supplement 1E*, S2 cells were transfected with P*actin::kpc-1::HA or* P*actin:: kpc-1R136AR143A::HA* and for *Figure 6D*, with P*actin::dma-1::Myc* alone or with P*actin::kpc-1::HA*. 3 days after transfection, cells were collected from T25 tissue culture flasks and lysed in lysis buffer (1xPBS, 1% Triton X-100 and 1% protease inhibitor cocktail (Sigma)) for 20 min on ice. Cell lysates were spun at 13,000 rpm for 10 min and supernatants were collected and detected using Western blot analysis with mouse antibody to HA (1:1000, Roche) or rabbit antibody to Myc (1:2000, Santa Cruz Biotechnology, Dallas, TX) and HRP-conjugated goat antibodies to mouse or rabbit (1:20,000, Jackson Immuno Research West Grove, PA).

For *Figure 4D*, twenty 10 cm NGM plates of each worm strain were lysed in RIPA lysis buffer (Sigma) with 1% protease inhibitor cocktail (Sigma) using Lysing Matrix C and the FastPrep-24 tissue homogenizer (MPbio, Santa Ana, CA). Lysates were spun at 13,000 rpm for 10 min. Supernatants

were collected and analyzed using Western blots with mouse antibody to GFP (1:1000, Roche, Indianapolis, IN), mouse antibody to FLAG (1:5000, Sigma) and HRP-conjugated goat antibody to mouse (1:20,000, Jackson Immuno Research).

## Biolayer interferometry

For protein expression, the ectodomain of DMA-1 (amino acids 20–507) was cloned into the baculoviral transfer plasmid pAcGP67-A (BD Biosciences Pharmingen, San Jose, CA) with a C-terminal hexahistidine tag, and the ectodomain of KPC-1 (amino acids 34–673) was cloned into pAcGP67-A with a C-terminal biotinylation-acceptor peptide and a hexahistidine tag. Proteins were expressed using the baculoviruses in High Five cells, and secreted into expression media (Insect-XPRESS, Lonza, Walkersville, MD, supplemented with 10 µg/mL gentamicin). Proteins were purified using affinity against $Ni^{2+}$-NTA agarose resin (Qiagen), followed by gel filtration chromatography with a Superdex 200 10/300 column in HEPES-buffered saline (HBS, 10 mM HEPES pH 7.2, 150 mM NaCl). Both proteins eluted as single, monodisperse peaks at volumes indicative of being monomeric. KPC-1 was further treated with the *E. coli* enzyme BirA (biotin ligase) for biotinylation, and re-purified with gel filtration chromatography.

We measured binding between KPC-1 and DMA-1 using biolayer interferometry (BLItz System, ForteBio, Pall Life Sciences, Port Washington, NY) with streptavidin tips (SA Biosensors; Fortebio, Menlo Park, CA) to capture biotinylated DMA-1. DMA-1 was immobilized to saturation at ~4.6 nm BLI responses. KPC-1 in HBS at varying concentrations from 0.45 µM to 218 µM was titrated on the biosensor tip. Association and dissociation kinetics were observed to be fast; equilibrium binding responses were used to fit the data to the one-site Langmuir absorption isotherm model (*Figure 7F*, blue circles). No corrections were applied for non-specific binding. The maximum response calculated from the isotherm (2.0 nm) indicates a biosensor surface with ~60% active protein, an experimentally sensible value. Dissociation constant ($K_D$) was calculated to be $33.5 \pm 5.3$ µM and $31.6 \pm 7.2$ µM over duplicate runs.

As a negative control, the SA tip was decorated with biotinylated ectodomain of human GPR56 (a kind gift from Celia Giulietta Fernandez) to ~11.5 nm BLI response levels. DMA-1 was observed not to bind to GPR56 (*Figure 7F*, open rectangles).

## Acknowledgements

This work was supported by the Howard Hughes Medical Institute and by NIH (1R01NS082208-01A1) to KS, by NIH (5R01GM111320) and by NSF (IOS-1455758) to C Chang. The work in the Özkan's lab was supported by a Klingenstein-Simons Fellowship in the Neuroscience. We thank the *Caenorhabditis* Genetics Center for the *kpc-1(gk8)* strain, the laboratories of L Luo and M Scott for *Drosophila* S2 and S2R+ cell lines, CG Fernandez for the GPR56 protein, and M Snyder for allowing us to use the FastPrep equipment. We also thank C Richardson for critical reading of the manuscript.

## Additional information

### Competing interests
KS: Reviewing editor, *eLife*. The other authors declare that no competing interests exist.

### Funding

| Funder | Grant reference number | Author |
| --- | --- | --- |
| Howard Hughes Medical Institute | | Xintong Dong<br>Wei Zou<br>Kang Shen |
| National Institutes of Health | 1R01NS082208-01A1 | Xintong Dong<br>Wei Zou<br>Kang Shen |
| National Institutes of Health | 5R01GM111320 | Hui Chiu<br>Yan Zou<br>Chieh Chang |

| National Science Foundation | Hui Chiu |
| | Yan Zou |
| | Chieh Chang |
| Esther A. and Joseph Klingenstein Fund | Yeonhee Jenny Park |
| | Engin Özkan |

The funders had no role in study design, data collection and interpretation, or the decision to submit the work for publication.

### Author contributions

XD, Conception and design, Acquisition of data, Analysis and interpretation of data, Drafting or revising the article; HC, YZ, Acquisition of data, Contributed unpublished essential data or reagents; YJP, EÖ, Designed and performed the experiments shown in Figure 7F, Acquisition of data, Analysis and interpretation of data, Drafting or revising the article; WZ, Drafting or revising the article, Contributed unpublished essential data or reagents; CC, Conception and design, Drafting or revising the article, Contributed unpublished essential data or reagents; KS, Conception and design, Analysis and interpretation of data, Drafting or revising the article

## Additional files

### Supplementary files

• Supplementary file 1. Table S1 Mutant alleles and transgenes used in this study. Table S2 Plasmids used in this study. Table S3 sgRNAs and repair oligos for CRISPR.

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
