## [Decision Letter]

Thank you for submitting your work entitled "Precise regulation of the guidance receptor DMA-1 by KPC-1/Furin instructs dendritic branching decisions" for peer review at *eLife*. Your submission has been favorably evaluated by Gary Westbrook (Senior editor) and three reviewers, one of whom, Hugo Bellen, is a member of our Board of Reviewing Editors.

The reviewers have discussed the reviews with one another and the Reviewing editor has drafted this decision to help you prepare a revised submission.

Summary:

*C. elegans* furin protein KPC-1 has recently been shown to regulate dendrite morphogenesis in multiple sensory neurons, including PVD, as well as processing insulins. Dong et al. now show that KPC-1 regulates the levels of the LRR receptor DMA-1 via late endosomal pathway to guide PVD dendrites. Although their findings that *kpc-1* acts cell autonomously at L3 stage for PVD dendrite morphogenesis are consistent with the published work by Salzberg et al., they present a different interpretation for the mutant phenotype, described as dendrite trapping, due to high signaling of the DMA-1 receptor in *kpc-1* mutants. Supporting their conclusion, they show that *kpc-1* mutants increase expression levels of DMA-1 and overexpression of *kpc-1* can prevent DMA-1 from interacting with its ligands SAX-7 and MNR-1. DMA-1 can be found in RAB-7 containing endosomes in PVD neurons. *kpc-1* appears to colocalize with RAB-7 in S2 cells, and promotes DMA-1 localization to late endosomes in PVD. Yet, the regulation of DMA-1 by KPC-1 does not require the intracellular domain of DMA-1. Overall, the data linking *kpc-1* to *dma-1* are strong, and offer new insights to dendrite morphogenesis and regulation of DMA-1. An issue is that the temporal regulation of DMA-1 by *kpc-1* during PVD dendrite outgrowth is mainly inferred through transgenic overexpression studies. The presentation of some data needs further clarification. Several figures lack quantification or controls.

Essential revisions:

An issue that was raised by two reviewers relates to the model that you propose. The work lacks any mechanistic insights into how KPC-1 targets DMA-1 to endosomes/lysosomes; and it does not convincingly show that this is the correct model. We find out that this targeting cannot be demonstrated to occur through cleavage of DMA-1 by KPC-1, which would be the obvious model. But this is negative evidence based on the lack of a detectable DMA-1 cleavage product. But, no experiments are done to prove that KPC-1 does not cleave DMA-1, or to validate the idea that KPC-1 might bind directly to DMA-1 without cleaving it. The lack of a detectable DMA-1 cleavage product might occur because the product is highly unstable.

Furthermore, the idea that the regulation of DMA-1 does not involve cleavage seems inconsistent with the finding that mutating the self-cleavage sequence in KPC-1 in a rescue construct (which we would assume is necessary to activate a preprotein to make an active protease?) rescues the null phenotype to a partial loss-of-function phenotype.

In addition, it should be relatively simple to use the tagged proteins that have been made, express them in S2 cells and show that they physically interact. Obviously, if the proteins are internalized and degraded too quickly, this may not be feasible. A time course experiment may then be needed. Alternatively, If the interaction is too weak, the method developed by Chris Garcia and colleagues on their campus could be used.

In the subsection “Partial loss of KPC-1 causes defects in self-avoidance of 3° branches and outgrowth of 4° branches” and Figure 3: the *kpc-1(xr58*) partial loss of function mutation led them to test the idea of incomplete cleavage of *kpc-1*. They show that KPC-1 expressed in S2 cells can be cleaved. They should test if KPC-1(R136A) blocks this cleavage.

In the last paragraph of the subsection “DMA-1 receptor level is increased in kpc-1 mutants” and Figure 5, Figure 7: the genetic analysis on *wy908* allele is a nice addition. It would be good to see if cytoplasmic deleted *dma-1* is still targeted to late endosomes.

---

## [Author Response]

*An issue that was raised by two reviewers relates to the model that you propose. The work lacks any mechanistic insights into how KPC-1 targets DMA-1 to endosomes/lysosomes; and it does not convincingly show that this is the correct model. We find out that this targeting cannot be demonstrated to occur through cleavage of DMA-1 by KPC-1, which would be the obvious model. But this is negative evidence based on the lack of a detectable DMA-1 cleavage product. But, no experiments are done to prove that KPC-1 does not cleave DMA-1, or to validate the idea that KPC-1 might bind directly to DMA-1 without cleaving it. The lack of a detectable DMA-1 cleavage product might occur because the product is highly unstable. In addition, it should be relatively simple to use the tagged proteins that have been made, express them in S2 cells and show that they physically interact. Obviously, if the proteins are internalized and degraded too quickly, this may not be feasible. A time course experiment may then be needed. Alternatively, If the interaction is too weak, the method developed by Chris Garcia and colleagues on their campus could be used.*

We thank the reviewers for these critical suggestions. We have collaborated with Dr Engin Özkan, former postdoc in the Garcia lab and now assistant professor at the University of Chicago, to specifically test the hypothesis that KPC-1 may target DMA-1 for degradation through directly binding to its ectodomain, a mechanism similar to how PCSK9 down-regulates LDLR. We have successfully expressed and purified predicted extracellular domains of DMA-1 and KPC-1. We have detected direct interaction between these ectodomains using biolayer interferometry (Figure 7). The dissociation constant (KD) measured is 33μM, which explains why it has been difficult for us to detect this weak interaction using low amounts of non-purified proteins on Western blots.

We are also developing a new assay to study in detail the cell biological aspects of DMA-1 trafficking. In these experiments, we try to label the plasma membrane fraction of DMA-1 with Fluorogen activating protein (FAP) and follow its endocytosis with live cell imaging. Due to technical difficulties, we are still troubleshooting these experiments. Meanwhile, we show unambiguously *in vivo* that KPC-1, which interacts with the ectodomain of DMA-1, functions synergistically with other endocytic mechanisms that utilize the cytosolic domain of DMA-1 (Figure 7). We thus hope that we have provided sufficient evidence that KPC-1 down-regulates DMA-1 by targeting it to endocytic compartments.

Furthermore, the idea that the regulation of DMA-1 does not involve cleavage seems inconsistent with the finding that mutating the self-cleavage sequence in KPC-1 in a rescue construct (which we would assume is necessary to activate a preprotein to make an active protease?) rescues the null phenotype to a partial loss-of-function phenotype.

*In the subsection “Partial loss of KPC-1 causes defects in self-avoidance of 3° branches and outgrowth of 4° branches” and Figure 3: the kpc-1(xr58) partial loss of function mutation led them to test the idea of incomplete cleavage of kpc-1. They show that KPC-1 expressed in S2 cells can be cleaved. They should test if KPC-1(R136A) blocks this cleavage.*

We want to thank the reviewers for this insightful suggestion. We identified two self-cleavage sites very close to each other (Figure 3—figure supplement 1). Mutating both sites completely eliminates the self-cleavage product (Figure 3—figure supplement 1). We have introduced these two mutations in the endogenous *kpc-1* locus using CRISPR. This mutant shows a full loss-of-function *kpc-1* mutant phenotype in the PVD neuron (Figure 3—figure supplement 1). These results show that removal of the N-terminal Pro domain is required for the activation of KPC-1.

Furthermore, we made a Pro domain deleted KPC-1 construct and expressed it in the *kpc-1(gk8)* mutant. This truncated protein fully rescues the PVD morphology phenotype of *gk8* mutants. Together, these data strongly suggest that KPC-1 activates itself through autocleavage.

*In the last paragraph of the subsection “DMA-1 receptor level is increased in kpc-1 mutants” and Figure 5, Figure 7: the genetic analysis on wy908 allele is a nice addition. It would be good to see if cytoplasmic deleted dma-1 is still targeted to late endosomes.*

We again followed this constructive suggestion. We expressed DMA-1△cyto-GFP in PVD. Compared to full length DMA-1-GFP, DMA-1△cyto-GFP shows reduced localization to the endosomes and increased localization to the plasma membrane (Figure 7). Furthermore, when we crossed this marker into *kpc-1* mutants, we observed that GFP signal is almost exclusively on the plasma membrane and very little in endocytic vesicles. Together, these results seem to be consistent with a model that KPC-1 and other endocytic mechanisms function synergistically to down-regulate plasma membrane DMA-1. KPC-1 interacts with the extracellular domain of DMA-1, while other endocytic machineries function through the cytosolic domain of DMA-1.